# OpenReview forum: "AutoHijacker: Automatic Indirect Prompt Injection Against Black-box LLM Agents"
_ICLR.cc/2025/Conference — Submitted to ICLR 2025_

### Official Review · Reviewer_8Tqa · 2024-10-26

**Soundness:** 3
**Presentation:** 3
**Contribution:** 3
**Rating:** 6
**Confidence:** 4

**Summary:**

This work introduces AutoHijacker, an automated black-box indirect prompt injection attack. It leverages the concept of LLM-as-optimizers. Specifically, it introduces a batch-based optimization framework to handle sparse feedback and also leverages a trainable memory to enable the effective generation of indirect prompt injections without continuous querying. Experiments are done on two benchmarks.

**Strengths:**

- The work presents AutjoHijacker as an automated black-box indirect prompt injection attack, which bridges the current research gap.
- The work did a good work in presenting the challenge of sparse feedback in indirect prompt injection tasks., and solve it in a simple and reasonable way.
- The results are promising with improvement over existing attacks on several LLMs.

**Weaknesses:**

I didn't see major flaws in the work and think it would be a good contribution to the community. I only have some questions for the authors regarding the evaluated defenses:
- The author did a great job in including defenses from the benchmarks. But I'm still curious how some state-of-the-art defenses could work for the attack: for example, in the work [Yi et al.], they show their white-box defense can reduce indirect prompt injection attack to nearly zero. Would the attack also work for such kinds of LLMs (optimized for defending against indirect prompt injection attacks)?
- I would recommend the author when introducing the concept of LLM-as-optimizer, can explain a little bit more before jumping into the challenge of sparse feedback.

Minor:
- missing "." line 185

**Questions:**

See weakness.

---

### Official Review · Reviewer_8fKc · 2024-11-02

**Soundness:** 2
**Presentation:** 3
**Contribution:** 2
**Rating:** 3
**Confidence:** 4

**Summary:**

The paper introduces AutoHijacker, an automatic black-box prompt injection attack. Built on the concept of LLM-as-optimizers, AutoHijacker constructs an attack memory through batch-based optimization and selects the most effective prompt injection case during the attack. Experimental results show that AutoHijacker outperforms previous attacks in effectiveness.

**Strengths:**

1. The paper analyzes the limitations of previous LLM-as-optimizers-based methods and proposes improvements to address them.

2. The proposed attack is black-box, making it applicable to certain closed-source LLMs, and therefore more broadly applicable than white-box attacks.

3. Experiments are conducted on two different benchmarks, comparing the effectiveness of various attacks.

**Weaknesses:**

1. The contributions of the paper appear to be incremental.

2. The improvement in the results does not seem significant, especially in comparison to the combined attack.

3. The paper lacks evaluation against effective defenses.

**Questions:**

1. The overall idea of the paper does not appear to be novel. The core concept still revolves around LLM-as-optimizers, which uses LLM responses to optimize attack prompts. This makes the paper's contribution seem somewhat incremental.

2. The evaluation results need further refinement. The paper describes the “combined attack” as a grey-box attack, but in practice, it’s often easy to know the purpose of an LLM application (especially for task-specific LLMs) and craft fake answers accordingly. Constructing a "combined attack" requires no optimization, which is much more efficient than AutoHijacker. Notably, the paper mentions a log length of 30, implying that a successful AutoHijacker attack requires at least 30 optimization iterations. Yet, the results show that AutoHijacker only achieves comparable performance to the combined attack. This suggests that the proposed attack is significantly less efficient.

3. The authors consider various defenses in Table 3, yet these defenses have been shown to be relatively ineffective in [1]. Why not test your attack against more robust defenses, such as Known-Answer Detection [1] or StruQ [2]?

[1] Formalizing and Benchmarking Prompt Injection Attacks and Defenses

[2] StruQ: Defending Against Prompt Injection with Structured Queries

4. I recommend including visual examples of AutoHijacker attacks to make the paper easier to understand. For instance, illustrations of specific attack strategies and guides used in the first step, "Meta Prompt Generation," would be helpful.

---

### Official Review · Reviewer_6Gim · 2024-11-02

**Soundness:** 3
**Presentation:** 2
**Contribution:** 2
**Rating:** 5
**Confidence:** 4

**Summary:**

The paper proposes a black-box prompt injection method that leverages LLMs as optimizers to inject prompts indirectly into LLM agents, utilizing minimal feedback and a trainable memory framework.

**Strengths:**

1. The batch-based optimization moves beyond single-injection attacks by utilizing multiple, diverse data to perform batch-based optimization, effectively addressing the sparse feedback issue that typically limits indirect prompt injection attacks.
2. The method shows state-of-the-art performance across multiple benchmarks, surpassing other attacks, and demonstrates high success on a real-world LLM agent.

**Weaknesses:**

1. Text and images need a better presentation. "Epochs" in figures need improvement for better readability. Terms like Mi,n, Di,n, Si,n are inconsistent which detracts from understanding.
2. The paper could further explore the use of diverse victim LLMs within the optimization process, examining how this might impact transferability across models or scales. Does the size or type of this victim LLM affect the overall results?

**Questions:**

1. When constructing N training data points, does the study explore the success probability of attacks in relation to different attack goals, variations in external data, and user instructions? Could the testing phase generate specific attack targets based on different query types and attack categories?
2. How does the scorer LLM contribute to optimization performance, and could its role be discussed in more detail?
3. What is the source and collection methodology for the meta prompts used in the training process?
4. How do the hyperparameters ktop and kbottom affect model performance, and could a more thorough analysis of these parameters improve the method's robustness?

---

### Official Review · Reviewer_rgkp · 2024-11-04

**Soundness:** 3
**Presentation:** 3
**Contribution:** 2
**Rating:** 3
**Confidence:** 2

**Summary:**

In this paper, the authors propose autohijacker, an automatic indirect black-box prompt injection attack. The results on two benchmark datasets indicate that it can be effective to both open-source and closed source models.

**Strengths:**

1 This paper is easy to follow.

2 The experiments are quite solid.

3 The soundness of the proposed method is good.

**Weaknesses:**

1 My biggest concern is the novelty of the proposed method. Although in Table 1 and Table 2, the results indicate that AutoJacker can achieve outstanding performances. However, the technical contribution only include a batch-based optimization framework and a trainable memory. It is a little marginal to me. However, I am open to this problem and delighted to further discuss with authors and other reviewers.

2 Details of the baseline attacks are needed. As far as I know, baseline methods such as PAIR are sensitive to various settings. Therefore, more details are required to provide to demonstrate the comparison is fair.

**Questions:**

1 Autohijacker is composed of two stages, including a training stage and a test stage. Therefore, my questions is how the authors divide the training data and the test data in their experiments.

2 Autohijacker needs three assistant LLMs, including a prompter, and attacker and a scorer. My question is how to choose those models in authors' experiments. Will stronger attacker bring higher ASR?

3 The authors show that AutoJacker can attack GPT-4o. How about other models such as Claude and Gemini?

---

### Meta-Review · Area_Chair_4WRi · 2024-12-19

**Metareview:**

This paper received three negative review and one positive review. The main concerns of reviewers are limited novelty, more details of baselines, more evaluation of defense, etc. However, the authors did not rebuttal so there is no discussion and further comments. After reading the paper and all reviews, the AC thinks the current version is still not ready for publication.

**Additional Comments On Reviewer Discussion:**

There is not rebuttal.

---

### Decision · Program_Chairs · 2025-01-22

Reject